# The Newborn Screening Programme Revisited: An Expert Opinion on the Challenges of Rett Syndrome

**DOI:** 10.3390/genes15121570

**Published:** 2024-12-05

**Authors:** Jatinder Singh, Paramala Santosh

**Affiliations:** 1Department of Child and Adolescent Psychiatry, Institute of Psychiatry, Psychology and Neuroscience, King’s College London, London SE5 8AF, UK; jatinder.singh@kcl.ac.uk; 2Centre for Interventional Paediatric Psychopharmacology and Rare Diseases (CIPPRD), South London and Maudsley NHS Foundation Trust, London SE5 8AZ, UK; 3Centre for Interventional Paediatric Psychopharmacology (CIPP) Rett Centre, Institute of Psychiatry, Psychology and Neuroscience, King’s College London, London SE5 8AF, UK

**Keywords:** newborn screening, Generation Study, Rett syndrome, presymptomatic testing, digital phenotyping

## Abstract

Genomic sequencing has the potential to revolutionise newborn screening (NBS) programmes. In 2024, Genomics England began to recruit for the Generation Study (GS), which uses whole genome sequencing (WGS) to detect genetic changes in 500 genes in more than 200 rare conditions. Ultimately, its purpose is to facilitate the earlier identification of rare conditions and thereby improve health-related outcomes for individuals. The adoption of rare conditions into the GS was guided by four criteria: (1) the gene causing the condition can be reliably detected; (2) if undiagnosed, the rare condition would have a serious impact; (3) early or presymptomatic testing would substantially improve outcomes; and (4) interventions for conditions screened are accessible to all. Rett syndrome (RTT, OMIM 312750), a paediatric neurodevelopment disorder, was not included in the list of rare conditions in the GS. In this opinion article, we revisit the GS and discuss RTT from the perspective of these four criteria. We begin with an introduction to the GS and then summarise key points about the four principles, presenting challenges and opportunities for individuals with RTT. We provide insight into how data could be collected during the presymptomatic phase, which could facilitate early diagnosis and improve our understanding of the prodromal stage of RTT. Although many features of RTT present a departure from criteria adopted by the GS, advances in RTT research, combined with advocacy from parent-based organisations, could facilitate its entry into future newborn screening programmes.

## 1. Background

Genomic sequencing programmes have the potential to facilitate newborn screening (NBS) by enabling access to treatment earlier at the population level, thereby improving longer-term outcomes for those affected. Globally, there are nine NBS programmes with sample sizes ranging from 1000 to 100,000 newborns [1]. Incorporating genomics into healthcare is a lengthy process dependent on various factors, including government engagement, cost-effectiveness, and other legislative issues [2]. Others have indicated a lack of robust evidence for introducing genomic sequencing into newborn screening [3]. Different factors will influence the landscape surrounding newborn screening programmes. From a capacity perspective, analysis and interpretation of bioinformatic and sequencing results will require trained professionals and educated healthcare providers with sufficient expertise. There will be a need for significant technological resources for data storage that, in turn, will need to comply with challenges to data privacy and security compliance laws when handling genomic data [4]. While data from the NBS programme BabySeq project has revealed clinically actionable findings, they have also highlighted challenges in translating these findings into routine health care and disseminating the actionability of genetic information for families [5]. Moreover, complexities surrounding consent should also be considered, and important questions have been raised regarding appropriate consent in genomic newborn screening. This includes offering families flexible options for data use and being able to re-assess results and recontact families, especially when variants of unknown significance are initially reported [6]. Despite these hurdles, initiatives such as the International Consortium on Newborn Sequencing (ICONS) are helping exchange best practices among NBS programmes [7].

In 2024, Genomics England began recruiting for the Generation Study (GS), which intends to use whole genome sequencing (WGS) to analyse the genetic codes of 100,000 newborn babies born in the UK. Specifically, it will examine whether WGS can facilitate the earlier detection of rare genetic conditions to allow for more timely interventions and improvements in quality of life [8]. The GS extends from the current newborn blood spot test offered to newborn babies in the UK, assessing nine rare conditions [9]. Currently, the GS uses WGS to detect genetic changes in about 500 genes in more than 200 rare conditions [10]. The conditions were selected based on four principles, including that (A) the gene causing the condition can be reliably detected, (B) the rare condition would have a harmful impact on the child if undiagnosed, (C) early or presymptomatic intervention would improve the child’s quality of life, and (D) an early intervention is equitable to all in the NHS [11]. As part of this process, more than 900 genes and associated conditions were reviewed to determine whether all four equally weighted principles could be satisfied. This opinion revisits the four principles of the GS from the perspective of Rett syndrome (RTT; OMIM 312750), providing evidence for each principle where necessary and bringing together essential information on the disorder’s complexity that could help inform policy-makers.

### Rett Syndrome

Rett syndrome is a lifelong neurological disorder with impairments across multiple organ systems. The disorder is usually caused by a de novo mutation in the *methyl CpG-binding protein 2 gene* (*MECP2*), with more than 500 different mutations identified [12]. There is a seemingly quiescent phase before neurological regression occurs and the emergence of specific problems such as autonomic dysregulation, epilepsy, motor impairments, cardio-respiratory abnormalities, and increased risk of sudden death [13]. Classical RTT is typified by a period of developmental regression and stereotypic hand movements. Not all individuals have these features, and those that do not meet all the main or supportive criteria are categorised as having atypical RTT. Nevertheless, about 95% of classical and 75% of atypical cases are said to have a pathogenic *MECP2* mutation [14]. Recent evidence from the US Natural History study has shown a higher proportion of hospitalisation in those with classic RTT than in those with atypical RTT [15]. However, RTT has heterogeneity on multiple levels, from the gene through to the symptom level, and variability in its presentation has been found across different geographical regions [16]. Indeed, the diagnosis of RTT is independent of its mutational profile and is based on clinical diagnostic criteria [17]. This presents diagnostic challenges because many patients do not fulfil the complete list of clinical diagnostic criteria.

## 2. The Gene Causing the Condition Can Be Reliably Detected

### 2.1. Genetic Prognosticators of RTT

At present, there is no cure for RTT, and the condition was first identified in 1966 in young females. Given its multisystem phenotype, RTT presents with a myriad of symptoms across both mental health and physical domains. However, unlike most neurological conditions, an animal model has shown that the condition is reversible even at the later stages of disease progression [18]. This suggests that the neuronal system in individuals with RTT is sufficiently intact to support functional restoration. Even though 95% of RTT patients with the classical RTT phenotype have a pathogenic *MECP2* mutation, some individuals with pathogenic *MECP2* mutations lack the clinical characteristics of RTT [19]. Genetic prognosticators of RTT also do not readily align with the clinical phenotype [20]. Likely pathogenic variants in *MECP2* have also been shown in girls without any overt neurodevelopmental symptoms [21]. All 200 conditions screened in the GS can be confirmed by an orthogonal assay [22], and in RTT there are likely to be issues with data interpretation, such as false positives. This aspect is important, as data suggest that newborns with a false positive screen reported higher healthcare use, i.e., hospitalisations, than controls [23]. Extrapolating evidence from 48 metabolic diseases [24], others have estimated that > 11,000 babies could have false positive results per year in the UK [3]. Understanding the natural history of the disorder and an evolving knowledge base could help mitigate problems in data interpretation. However, as noted by others, this interpretation is likely to be impacted by limitations in genetic databases because most genomic studies (86.3%) have been performed in individuals of European ancestry [25]. This could, in theory, skew the frequency of false negatives/false positives because of the limited knowledge of disease-causing RTT genes in underrepresented populations and exacerbate existing health disparities. Roadmaps for incorporating diverse populations in genomics research have been developed to address imbalances in genomic studies [25], which may facilitate data interpretation.

### 2.2. Genetic Modifiers

Even when there is a combination of genetic testing and clinical assessment, this may only lead to a diagnosis in about half of patients [26]. In these situations, unusual symptoms could make precise diagnosis more challenging because it would be difficult to determine whether an unusual symptom represents a novel phenotype or is due to a secondary disease. Sometimes colloquially known as “double trouble” [27,28], cases of double trouble have been reported in the literature and suggested to be more prevalent than initially thought [28,29]. Double trouble disease phenotypes are more likely to result from two genes with similar pathways [29]. There is emerging evidence that RTT may also have a double trouble disease phenotype. Our recent work has shown that co-occurring single nucleotide polymorphisms (SNPs) are modifiers of clinical severity in RTT, most likely interacting through methylation pathways [30,31]. Some other evidence has also suggested that copy number variants, more so than SNPs, can act as genetic modifiers that alter the clinical phenotype in individuals with RTT [32]. Next-generation sequence methods may not be able to detect additional diagnoses, especially methylation defects, and for RTT, the possibility of a dual diagnosis should be considered. From the perspective of the GS, newborns with RTT should not be categorised as ‘neuro-genomic’ events. Multiple events occur in a child’s epigenome during early life stages, and it would be prudent for epigenetic marks of RTT to be tracked over time.

## 3. The Rare Condition Would Have a Harmful Impact on the Child If Undiagnosed

### The Burden of RTT

This principle considered the impact on the child’s quality of life if the disorder was undiagnosed. Genomics England acknowledges that “[…] it can be challenging to determine what causes a ‘debilitating impact’ on the quality of life […]” [33], and recognises that it is vital to include evidence from those who have experience with the condition. In the UK, there is a dire need for services to improve the quality of life (QoL) of children, young people, and adults with RTT. There is a significant gulf between treatment and day-to-day expectations, and most individuals with RTT ultimately depend on 24 h care. This places an immutable burden on their families and diagnostic isolation. Early diagnosis of RTT would facilitate assessment, prevent misdiagnosis, allow timely access to physiotherapists and speech and language therapy, and thereby allow treatment strategies to be implemented before symptoms that are relatively straightforward to treat become entrenched, chronic, and severe. The burden of illness study showed the most impactful symptoms affecting QoL were those related to the core features of RTT [34]. RTT’s impact on quality of life can have many layers, affecting siblings [35] and caregivers [36,37]. This is not pathognomonic for RTT (for example, epilepsy in Dravet syndrome). Some evidence has shown that controlling seizure activity in individuals with RTT is an important factor in improving the QoL of parents [38]. However, even milder symptoms can have a disproportionate impact on caregivers [27]. Of note, episodic symptoms were also found to affect caregivers more than those with RTT [34]. The level of unmet need for managing and treating mental health and physical health problems within the RTT community is high. If left undiagnosed, it is impossible to overstate the debilitating impact on individuals with RTT and their families and the major burden this places upon health services and wider society.

## 4. Early or Presymptomatic Intervention Would Improve the Child’s Quality of Life

### Early Presymptomatic Testing

At present, the GS includes disorders for which interventions could be initiated in early childhood before the child shows signs of illness that could modify or delay the condition. The assumption that individuals with classical (typical) RTT have a seemingly normal developmental epoch during the first 6–18 months of life before regression is somewhat premature. A growing body of evidence suggests that early development is not normal in RTT [39]. We emphasise that the time window before regression represents a period of untapped opportunities for presymptomatic RTT testing. This is a burgeoning area of research, but faces different challenges. In 2021, a landmark study showed that presymptomatic training reduces functional impairments, particularly with motor and memory tasks, in a mouse model of RTT [40]. The authors go on to state that presymptomatic intervention, i.e., early behavioural training, for RTT could delay symptom onset and provide a justification for genetic screening for RTT in newborn babies [40,41]. This rationale also aligns with the premise that environmental enrichment for RTT patients as young as 3 years old has led to improved motor skills [42]. Regarding RTT, some data using MRI have revealed specific brain MRI patterns in girls aged 3.5 (mean age) years old [43], and recently, we have shown that biometric sensor-guided monitoring can detect symptom change at an individual patient level in early childhood [30]. However, data on early or presymptomatic intervention, and whether it would improve the child’s quality of life, are limited. There needs to be a concerted effort to develop detectable biomarkers. Many genetic disorders, as well as complex multigenic conditions, remain undetectable using current biochemical methods.

Unlike other neurodevelopmental disorders such as tuberous sclerosis complex (TSC), which also has multisystem involvement, there are currently no available (prenatal or presymptomatic) biomarkers for RTT to detect changes before the child shows signs of illness. In individuals with TSC, the condition can be detected prenatally based on the presence of cardiac rhabdomyomas or other TSC-associated lesions using foetal ultrasounds [44,45]. Moreover, EEG and MRI biomarkers can now reliably identify TSC in individuals during early childhood [46]. Notwithstanding this limitation, emerging evidence has indicated that the period before regression is not normal for RTT, with deficits occurring before the onset of obvious clinical symptoms [47]. It has been suggested regarding RTT that there may be abnormal movements, tremors, body stereotypes, and abnormal facial and finger movements during the first few months of life [48,49,50]. Linguistic impairments and delays in communicative gestures have also been noted during early developmental stages of RTT [51,52]. However, as mentioned by others, the wide variability of subtle developmental impairments observed in individuals with RTT during early life stages has stymied screening of early pathogenic signs [47]. Unless strategies are adopted for early surveillance for RTT, the scarcity of data collection during the presymptomatic phase hinders progress in individuals with RTT, which could potentially otherwise lead to an earlier diagnosis and better understanding of presymptomatic profiles.

## 5. Early Intervention Is Equitable and Accessible to All

### 5.1. Early Intervention

This principle relates to conditions where there is already a care and support system within the NHS for the child and family. Rett syndrome management is happening piecemeal, and individuals are passed around from one specialist to another. Care depends upon a multidisciplinary team, and an individualized approach is required in nearly all cases. Part-funded by the parent-based charity Reverse Rett, in 2019, the Centre for Interventional Paediatric Psychopharmacology (CIPP) Rett Centre was established at the Maudsley Hospital, London and King’s College London. The CIPP Rett Centre is a national service that provides multidisciplinary care and precision medicine strategies for infants, young people, and adults with RTT in the UK and EU. It focuses on personalised medicine combined with innovative predictive measures to risk stratify patients with RTT who are at the most risk. These include individuals with complex symptoms across different organ systems that need timely interventions. The centre specialises in emotional, behavioural, and autonomic dysregulation (EBAD), including respiratory issues and personalised care using pharmacogenomics and sensor-based physiological data. The CIPP Rett Centre allows clinical research to be translated proficiently into routine clinical care, as a model for rare disease care.

At present, there is no specific intervention for RTT in the UK. In 2023, trofinetide was approved in the US and is the only FDA drug for use in patients with RTT aged two and over [53]. Trofinetide is derived from tripeptide glycine-proline-glutamic acid (GPE), which is the naturally occurring form of insulin-like growth factor 1 [54]. Previous evidence using recombinant IGF-1 for RTT has had mixed results [55]. However, trofinetide may have pharmacokinetically unique properties compared to endogenous GPE, resulting in improved oral bioavailability [54]. The exact mechanism of action of trofinetide remains unknown, but it is thought to exert its action through IGF-1 to modulate neuroplasticity and dampen neuroinflammation [56]. Trofinetide may modulate neuroplasticity and neuroinflammation by interacting with downstream signalling cascades associated with IGF-1 activation. This, in turn, is thought to increase the expression of synapse-forming proteins [54], which could improve synaptic function and architecture [56]. Overall, it has been suggested that trofinetide mirrors the neuroprotective effects of GPE by reducing neuroinflammation and excitotoxity [56]. While the approval of trofinetide presents a significant milestone in RTT drug research, it is not a cure for RTT, and the way forward in the UK is unknown. Given the multiple organ systems involved in RTT, trofinetide will likely be used as an adjunct alongside other medications that help manage the complexity of symptoms.

### 5.2. Transformative Therapies

Transformative therapies, particularly gene delivery methods of healthy *MECP2* copies into mutated cells, offer another opportunity for intervention [57,58,59]. Our evidence synthesis suggests that gene therapy could be an attractive proposition for RTT treatment, and we hypothesise that early intervention is better for preventing symptoms from manifesting [60]. While gene therapy for RTT can potentially address the disorder’s root cause mostly in those with classical RTT, aspects regarding affordability, uncertainty, and infrastructure requirements should also be considered [61]. Gene therapy for RTT is most likely to be delivered as a one-off intervention, and payers would need to consider the high upfront cost and the longer-term benefits it brings. Recently, the National Institute for Health and Care Excellence (NICE) rejected the Alzheimer’s drug donanemab due to uncertainties in its health economics modelling, including the longer-term effects of donanemab and related treatment costs [62]. The long-term safety effects of gene therapy products should not be understated. A recent study reported haematological cancer observed in seven cases post lentiviral gene therapy for cerebral adrenoleukodystrophy [63]. While current trials examine the safety, tolerability, and preliminary efficacy of gene therapy for RTT, the effectiveness of transgenes might be uncertain, especially during the early stages before pivotal longer-term efficacy trials begin. As transformative therapies use advanced technology and are resource intensive, there also needs to be sufficient infrastructure in place to identify suitable patients, deliver the gene therapy, and monitor health-related QoL outcomes longitudinally. Long-lasting expression of the transgene, combined with the appropriate dose, rigorous safety monitoring, infrastructure requirements, and measuring the health-related QoL of individuals with RTT and their caregivers, will be crucial to consider when developing interventions that are equally accessible to all.

## 6. Possible Future Directions

Digital phenotyping combined with machine learning could be used to develop prediction models that help to detect early signs of RTT. The potential for digital phenotyping to assess phenotypes before any symptoms of the disease are known, and the ability to generate large datasets of relatively uncommon phenotypic features implemented more widely across diverse populations at low cost [64], are an ideal foil to examine the prodromal stages of RTT. Some studies have indeed used digital phenotyping to separate prodromal states when compared to controls in other neurological diseases that share overlapping symptoms with RTT, such as saccades and finger stereotypies in Huntington’s disease [65,66], gait impairments in spinocerebellar ataxia [67], and prodromal speech biomarkers in Parkinson’s disease [68]. Artificial intelligence could facilitate precise diagnosis based on facial gestures and communication. Digital phenotyping has been used as a predictor for mental health based on images [69], while speech analysis can predict mental health changes in emergent psychosis [70]. A digital phenotyping approach for RTT needs to be scalable, and emerging evidence from studies of autism spectrum disorder (ASD) has shown promise. Recently, an algorithm that combines multiple digital phenotypes and digital behavioural phenotyping has allowed for the early detection of ASD with high diagnostic accuracy [71]. This finding adds weight to the premise that quantifying multiple RTT-like behaviours could lead to the development of prediction models that would allow the early detection of RTT-like behaviours before overt symptoms appear, even when there is variable presentation. Such an approach could also be helpful when combined with passive data collection from biometric sensors, especially given that events during presymptomatic stages are likely episodic. Here, data from biometric sensors that help detect pre-ictal events [72], combined with motor function, saccades, speech, and facial expression data, could further assist presymptomatic testing.

Artificial intelligence could lead to quasi-causal inferences from data. This approach might also uncover episodic events that could allow clinicians to provide time-specific interventions and thereby mitigate or even delay overt symptoms from manifesting [47]. Such a strategy could reshape how those with RTT are cared for in the UK and ultimately improve their health-related QoL. Unlike some other neurodevelopmental disorders, the number of individuals with RTT is rare [73], and models would be trained on relatively small datasets. As mentioned by others [64], this raises the question of what evolutionary stage of digital phenotyping would be enough to distil into clinical practice and provide new actionable disease insights that would enrich the evidence base for other diseases to be considered for the GS. Establishing RTT centres of excellence to enrich the current body of evidence, especially focusing on using digital phenotyping to examine the prodromal stages of RTT and using relevant genetic methods to probe for a second condition, could help RTT be incorporated into the next phase.

## 7. Conclusions

The GS will make a transformational difference to the current ecosystem for children with rare diseases. We believe associative genetic modifiers should be examined in RTT patients, especially when dual genes interact with similar pathways to those involved in gene methylation. However, these genes could also operate, in part, via a methylation-independent mechanism [74]. We also raise the importance of using digital phenotyping to examine the phenotype of individuals before there are any signs of disease. Interrogating large data sets and returning inferences applicable to an individual child could be a boon for patients with RTT (Figure 1). This could be a digital archive of videos/images of individuals with RTT from birth that would be sufficiently powered to detect change and then compared to the developmental milestones of neurotypical children.

In the US, data suggest that it takes, on average, 9.5 years for a new condition to be adopted across all states [75]. Including diseases without a recognised biomarker into newborn screening programmes has been debated [22]. Indeed, the US observational NBS studies ‘Early Check’ and ‘Guardian’ also have a panel of early onset disorders for which no treatment is available [6]. This represents a departure from criteria adopted by the GS, but does provide scope for change in NBS programmes. As the GS evolves, advances in biomarker development, better understanding of genotype–phenotype associations in RTT research, and advocacy from parent-based organisations could hasten RTT’s entry into newborn screening programmes.

## Figures and Tables

**Figure 1 genes-15-01570-f001:**
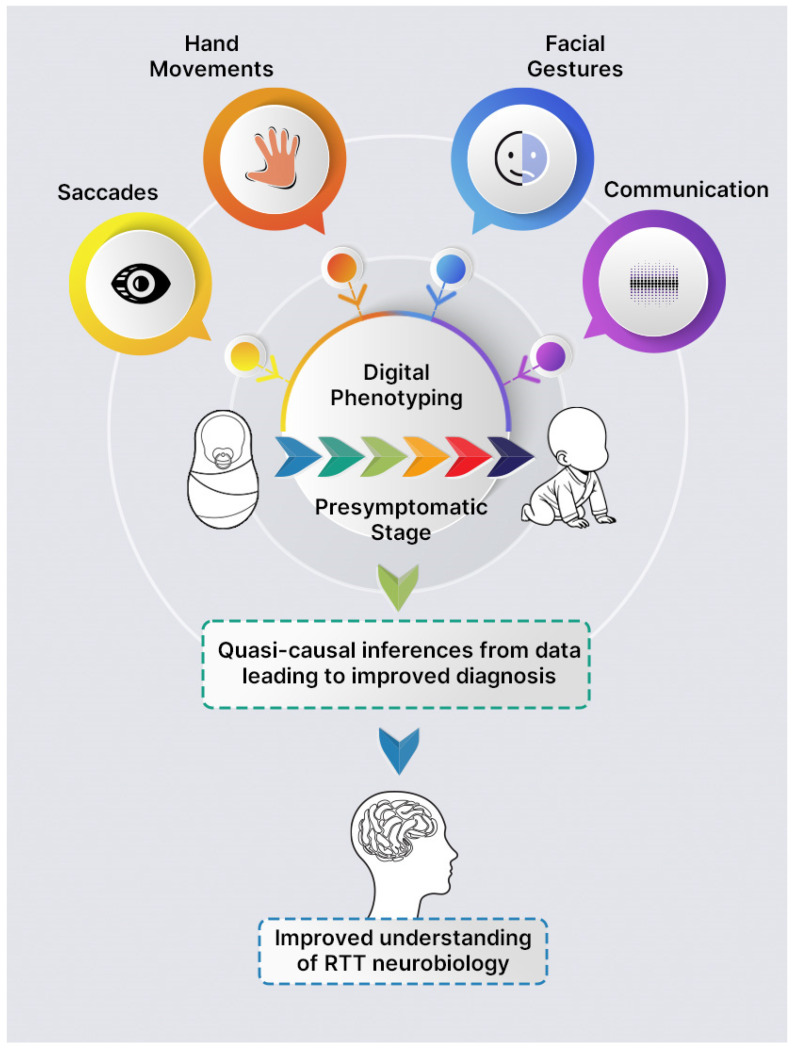
Digital phenotyping of individuals with Rett syndrome. Abbreviations: RTT, Rett syndrome. Notes: Digital phenotyping in individuals with Rett syndrome (RTT) could facilitate diagnosis before the child shows signs of illness (in the presymptomatic stage). Interrogating a digital repository of images/videos from birth for hand movements, saccades, facial gestures, and speech combined with the predicted power of artificial intelligence could lead to quasi-causal inferences from data. This would improve our understanding of RTT neurobiology, improve diagnostic accuracy, and allow for early intervention that could modify or delay symptoms from emerging.

## Data Availability

Evidence presented in this article is freely available.

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
