# Peer review of "The Newborn Screening Programme Revisited: An Expert Opinion on the Challenges of Rett Syndrome"

_genes, 2024, doi:10.3390/genes15121570_

Round 1
Reviewer 1 Report
Comments and Suggestions for Authors
In this opinion article, the authors revisit the GS and discuss Rett syndrome (RTT) from the perspective of the four criteria indicated by GS. They suggest that although many features of RTT present a departure from the GS criteria, advances in RTT research could allow its future entry in newborn screening programmes.
The manuscript is well written and provide a clear pricture of the current knowledge about RTT and its prospective of therapy and early diagnosis.

Author Response
Reviewer 1:
In this opinion article, the authors revisit the GS and discuss Rett syndrome (RTT) from the perspective of the four criteria indicated by GS. They suggest that although many features of RTT present a departure from the GS criteria, advances in RTT research could allow its future entry in newborn screening programmes.
The manuscript is well written and provide a clear picture of the current knowledge about RTT and its prospective of therapy and early diagnosis.
Response: Thank you for your comments.

Reviewer 2 Report
Comments and Suggestions for Authors
The manuscript of Singh and Santosh is a comprehensive article on the newborn screening program in the context of Rett syndrome. The text is logically structured. First, the authors provide background on the newborn screening program and Rett syndrome clinics, highlighting heterogenicity. Especially worthoworthy are the recent advances in the context of Rett syndrome, such as genetic modifiers, diversity aspects in natural history studies, digital phenotypic and newest therapies. Altough simple, figure is illustrative.
This is a very important opinion.
Comments to authors:
1. Linne 35 should be nine NBS programmes in the UK
2. Please provide info on the mechanism of action of trofinetide
3. Conclusions could be shortened

Author Response
Reviewer 2
The manuscript of Singh and Santosh is a comprehensive article on the newborn screening program in the context of Rett syndrome. The text is logically structured. First, the authors provide background on the newborn screening program and Rett syndrome clinics, highlighting heterogenicity. Especially worth worthy are the recent advances in the context of Rett syndrome, such as genetic modifiers, diversity aspects in natural history studies, digital phenotypic and newest therapies. Although simple, figure is illustrative.
This is a very important opinion.
Comments to authors:
- Line 35 should be nine NBS programmes in the UK
- Please provide info on the mechanism of action of trofinetide
- Conclusions could be shortened
We thank the author for their review and address the minor comments below:
- Line 35 should be nine NBS programmes in the UK
Response: We thank the reviewer for pointing this out and it should have been nine NBS programmes globally and have updated the manuscript to reflect this change. The new text is:
Revised text:
“Globally, there are nine NBS programmes with sample sizes ranging from 1,000 to 100,000 newborns [1]”
- Please provide info on the mechanism of action of trofinetide
Response: While the exact mechanism of how trofinetide my modulate neuroplasticity and dampen neuroinflammation, we have expanded this section to state that, through IGF-1 activation, trofinetide may increase the expression of synapse-forming proteins that could improve synaptic function and architecture.
New text:
“Trofinetide may modulate neuroplasticity and neuroinflammation by interacting with downstream signalling cascades associated with IGF-1 activation. This in turn is thought to increase the expression of synapse forming proteins [47] that could improve synaptic function and architecture [49]. Overall, it has been suggested that trofinetide mirrors the neuroprotective effects of GPE by reducing neuroinflammation and excitotoxity [49].”
- Conclusions could be shortened
Response: Thank you for this suggestion. We have shortened the Conclusion as far as possible without losing key information. We have also added a new sentence with a reference. The revised conclusion has been shortened from 267 to 238 words. We hope that you will find this acceptable.

Reviewer 3 Report
Comments and Suggestions for Authors
The Newborn Screening Programme Revisited: Expert Opinion on the Challenges for Rett Syndrome by Drs. Jatinder Singh and Paramala Santosh is an opinion article well written and focused in a challenging and controversial theme. With their opinion, the authors revisit the GS (Generation Study from Genomics England in UK), discussing a putative future inclusion of RETT syndrome in the neonatal screening by reaching the four principles which the GS is based. However, some significant changes need to be addressed:
Major concerns that will be discussed:
.-Normally, decisions about which diseases are covered take long and may depend of National or Regional/Federal Government must be commented.
.-No major comments has been made regarding ethical concerns. Public Health Impact Assessment is strictly necessary to be addressed. In fact, The BabySeq Project in US, has shown that genomic screening in healthy neonates can reveal actionable information, though there is still debate over how best to integrate such data into routine care and communicate results to families in a responsible manner.
.- It is important to enable timely interventions; evaluating the psychosocial and economic impacts on families;
.-No major comments has been made regarding future or present regarding pharmacogenomics implementation. Integrate pharmacogenomics to optimize drug therapies to RTT, and establish personalized medication strategies from birth, advancing precision medicine for neonates and children must be commented
.-No major comments has been made regarding bioinformatics items to be considered, and follow-up care. In fact, maintain the technology infrastructure for data storage, analysis, and sharing after including new diseases. They would ensure that genome data is stored securely and in compliance with privacy laws.
.-Some comments regarding Informed Consent have missing. The authors must discuss offering families’ flexible options for data use and secondary findings.
.-I also missed some introducing background regarding such initiatives such as ICONS (International Consortium on Newborn Sequencing), while setting a benchmark for future healthcare advancements in precision medicine.
.-I agree that as genomic sequencing becomes more accessible and cost-effective, there is growing interest in newborn genome screening (NGS) both for identifying immediate health risks and understanding an individual’s genetic predispositions across their lifespan. In this context, NGS aims to address both acute and long-term health needs, reflecting a significant shift in RTT including approach to pediatric care. Guidelines for the use of genome screening in neonates are still being developed, and MS like is mentioned here are critical in informing future best practices. As more data becomes available, it will be essential to consider both the benefits and limitations of NGS, as well as the infrastructure needed to support its implementation at a population level.
.-Regarding this opinion: revisits the four principles of the GS from the perspective of Rett Syndrome, providing evidence for each principle were debatable and interest conflicting opinions:
i)The gene causing the condition can be reliably detected. As the authors commented Genetic prognosticators of RTT also do not readily align with the clinical phenotype, in addition in RTT, there are likely to be issues with data interpretation, such as false positives, or some other evidence has also suggested that copy number variants, more so than SNPs, can act as genetic modifiers that 108 alter the clinical phenotype in individuals with RTT.
ii)The rare condition would have a harmful impact on the child if gone undiagnosed. The impact of quality of life in RTT can have many layers, affecting siblings and caregivers. This is not pathognomic for RTT, there many Rare Diseases, eg suffering from epilepsy such as Dravet or WEST syndromes.
iii) Early or presymptomatic intervention would improve the child’s quality of life. Although some data pointed out to this aspect. In my opinion is not sufficient. One of the most critical aspects is to have biomarkers that allow this statement to be generated or discarded. However, while effective, biochemical assays are inherently limited by the availability of detectable biomarkers for each condition, as well as the specificity and sensitivity of each test. Many rare genetic disorders, as well as complex multigenic conditions, remain undetectable using current biochemical methods.
iv) Early intervention is equitable and accessible to all. Limitation is at present clearly, there is no specific intervention for RTT in the UK. On the other hand, I agree gene therapy could be an attractive proposition in RTT, not immediately.
Reducing on average the take a new condition to be included in the NB Screening across the countries is a real challenge, any thin, such this article debating the theme is really useful. I think that it is not a question of time, but of quality in the rational use of economic resources in healthcare, becoming a politic issue. Inclusión new diseases in the NBS has to be the result of a close agreement between many parties, based on economic availability but with a strong scientific basis.

Author Response
Reviewer 3
The Newborn Screening Programme Revisited: Expert Opinion on the Challenges for Rett Syndrome by Drs. Jatinder Singh and Paramala Santosh is an opinion article well written and focused in a challenging and controversial theme. With their opinion, the authors revisit the GS (Generation Study from Genomics England in UK), discussing a putative future inclusion of RETT syndrome in the neonatal screening by reaching the four principles which the GS is based. However, some significant changes need to be addressed:
Major concerns that will be discussed:
1, Normally, decisions about which diseases are covered take long and may depend of National or Regional/Federal Government must be commented.
2, No major comments has been made regarding ethical concerns. Public Health Impact Assessment is strictly necessary to be addressed. In fact, The BabySeq Project in US, has shown that genomic screening in healthy neonates can reveal actionable information, though there is still debate over how best to integrate such data into routine care and communicate results to families in a responsible manner.
3, It is important to enable timely interventions; evaluating the psychosocial and economic impacts on families;
4, No major comments has been made regarding future or present regarding pharmacogenomics implementation. Integrate pharmacogenomics to optimize drug therapies to RTT, and establish personalized medication strategies from birth, advancing precision medicine for neonates and children must be commented
5, No major comments has been made regarding bioinformatics items to be considered, and follow-up care. In fact, maintain the technology infrastructure for data storage, analysis, and sharing after including new diseases. They would ensure that genome data is stored securely and in compliance with privacy laws.
6, Some comments regarding Informed Consent have missing. The authors must discuss offering families’ flexible options for data use and secondary findings.
7, I also missed some introducing background regarding such initiatives such as ICONS (International Consortium on Newborn Sequencing), while setting a benchmark for future healthcare advancements in precision medicine.
Response: We thank the reviewer for raising these points. Except for #4, we have now added a new paragraph within the background section highlighting all these points in the revised manuscript and provide additional references:
#4 We have not included a comment on pharmacogenomics (PGx) implementation and how it maybe implemented to optimize drug therapies to RTT. This is because the evidence base for the use of PGx in children is limited with only a few studies conducted in young people [1]. Studies on paediatric PGx-guided prescribing are scarce, and three drugs (codeine, lansoprazole, and omeprazole) are suggested to have age-specific paediatric guidance [2]. Others have shown that that PGx-guided treatment could be helpful for mood disorders and gastritis/esophagitis in the paediatric population [3], however it is unclear how useful this would be for neonates. Hence based on the limited evidence, we feel that it perhaps premature at this stage to comment on the usefulness of PGx implementation and its feasibility in advancing precision medicine for neonates.
References
[1] Malik S. et al. Pharmacogenetics in Child and Adolescent Psychiatry: Background and Evidence-Based Clinical Applications. J Child Adolesc Psychopharmacol. 2024 Feb;34(1):4-20.
[2] Aka I., Bernal C.J., Carroll R., Maxwell-Horn A., Oshikoya K.A., Van Driest S.L. Clinical Pharmacogenetics of Cytochrome P450-Associated Drugs in Children. J. Pers. Med. 2017;7:14.
[3] Roberts T.A., Wagner J.A., Sandritter T., Black B.T., Gaedigk A., Stancil S.L. Retrospective Review of Pharmacogenetic Testing at an Academic Children’s Hospital. Clin. Transl. Sci. 2020;14:412–421.
New text:
Incorporating genomics into healthcare is a lengthy process and dependent on a variety of factors including government engagement, cost effectiveness and other legislative issues [2]. Others have indicated a lack of robust evidence for introducing genomic sequencing into newborn screening [3]. Different factors will influence the landscape surrounding newborn screening programmes. From a capacity perspective, analysis and interpretation of bioinformatic and sequencing results will require trained professionals and/or educated healthcare providers with sufficient expertise. There will be a need for significant technological resources for data storage that in turn will need to comply with challenges to do with data privacy and security compliance laws when handling genomic data [4]. While the data from the NBS programme BabySeq project has revealed clinically actionable findings it has also highlighted challenges in how best to translate these findings into routine health care and to disseminate the actionability of the genetic information for families [5]. Moreover, the complexities surrounding consent should also be considered and important questions have been raised in the use of appropriate consent in genomic newborn screening. This includes offering families’ flexible options for data use and being able to re-assess the results and recontact families especially when variants of unknown significance are initially reported [6]. Despite these hurdles, initiatives such as the International Consortium on Newborn Sequencing (ICONS) are helping to exchange best practice among NBS programmes [7].
References
[2] Stark Z, Dolman L, Manolio TA, Ozenberger B, Hill SL et al. Integrating Genomics into Healthcare: A Global Responsibility. Am J Hum Genet. 2019 Jan 3;104(1):13-20.
[3] Turnbull C, Firth HV, Wilkie AOM, Newman W, Raymond FL, et al. Population screening requires robust evidence-genomics is no exception. Lancet. 2024 Feb 10;403(10426):583-586.
[4] Jiang S, Wang H, Gu Y. Genome Sequencing for Newborn Screening-An Effective Approach for Tackling Rare Diseases. JAMA Netw Open. 2023 Sep 5;6(9):e2331141.
[5] Green RC, Shah N, Genetti CA, Yu T, Zettler B. et al. BabySeq Project Team. Actionability of unanticipated monogenic disease risks in newborn genomic screening: Findings from the BabySeq Project. Am J Hum Genet. 2023 Jul 6;110(7):1034-1045.
[6] Knoppers BM, Bonilha AE, Laberge AM, Ahmed A, Newson AJ. Genomic sequencing in newborn screening: balancing consent with the right of the asymptomatic at-risk child to be found. Eur. J Hum Genet. 2024 Aug 12.
[7] https://www.iconseq.org/ (accessed 27 November 2024)
8, I agree that as genomic sequencing becomes more accessible and cost-effective, there is growing interest in newborn genome screening (NGS) both for identifying immediate health risks and understanding an individual’s genetic predispositions across their lifespan. In this context, NGS aims to address both acute and long-term health needs, reflecting a significant shift in RTT including approach to pediatric care. Guidelines for the use of genome screening in neonates are still being developed, and MS like is mentioned here are critical in informing future best practices. As more data becomes available, it will be essential to consider both the benefits and limitations of NGS, as well as the infrastructure needed to support its implementation at a population level.
Response: Thank you for your comment.
Regarding this opinion: revisits the four principles of the GS from the perspective of Rett Syndrome, providing evidence for each principle were debatable and interest conflicting opinions:
i)The gene causing the condition can be reliably detected. As the authors commented Genetic prognosticators of RTT also do not readily align with the clinical phenotype, in addition in RTT, there are likely to be issues with data interpretation, such as false positives, or some other evidence has also suggested that copy number variants, more so than SNPs, can act as genetic modifiers that 108 alter the clinical phenotype in individuals with RTT.
Response: Thank you
ii)The rare condition would have a harmful impact on the child if gone undiagnosed. The impact of quality of life in RTT can have many layers, affecting siblings and caregivers. This is not pathognomic for RTT, there many Rare Diseases, eg suffering from epilepsy such as Dravet or WEST syndromes.
Response: We have added the aspect to with pathognomonic and given the example of Dravet syndrome.
iii) Early or presymptomatic intervention would improve the child’s quality of life. Although some data pointed out to this aspect. In my opinion is not sufficient. One of the most critical aspects is to have biomarkers that allow this statement to be generated or discarded. However, while effective, biochemical assays are inherently limited by the availability of detectable biomarkers for each condition, as well as the specificity and sensitivity of each test. Many rare genetic disorders, as well as complex multigenic conditions, remain undetectable using current biochemical methods.
Response: Thank you for pointing this out and we have added this text.
Early intervention is equitable and accessible to all. Limitation is at present clearly, there is no specific intervention for RTT in the UK. On the other hand, I agree gene therapy could be an attractive proposition in RTT, not immediately.
Response: We agree.
Reducing on average the take a new condition to be included in the NB Screening across the countries is a real challenge, any thin, such this article debating the theme is really useful. I think that it is not a question of time, but of quality in the rational use of economic resources in healthcare, becoming a politic issue. Inclusión new diseases in the NBS has to be the result of a close agreement between many parties, based on economic availability but with a strong scientific basis.
Response: Yes, we agree and have alluded to these points in the background section of the revised manuscript.
